# Lifestyle-Related Risk Factors for the Incidence and Progression of Chronic Kidney Disease in the Healthy Young and Middle-Aged Population

**DOI:** 10.3390/nu14183787

**Published:** 2022-09-14

**Authors:** Akihiro Kuma, Akihiko Kato

**Affiliations:** 1Kidney Center, Hospital of the University of Occupational and Environmental Health, 1-1 Iseigaoka, Yahatanishi-ku, Kitakyushu 807-8556, Fukuoka, Japan; 2Blood Purification Unit, Hamamatsu University Hospital, 1-20-1 Handayama, Higashi-ku, Hamamatsu 431-3125, Shizuoka, Japan

**Keywords:** occupational health, obesity, dyslipidemia, hyperuricemia, metabolic syndrome

## Abstract

The prevalence of chronic kidney disease (CKD) increased by 88% from 1990 to 2016. Age of onset of lifestyle-related diseases (such as hypertension, diabetes mellitus, obesity, dyslipidemia, and hyperuricemia), which are risk factors for incident CKD, is lower now compared with the past. Thus, we aimed to evaluate the risk factors for the incidence and progression of CKD in the young and middle-aged population. There are differences in the risk for CKD among the young, middle-aged, and elderly populations. We aimed to assess obesity (which is basic component of metabolic syndrome), waist circumference, and abdominal adiposity, which are predictive factors of CKD in the younger population. Furthermore, we described the management and clinical evidence of hypertension, diabetes mellitus, dyslipidemia, and hyperuricemia for young and middle-aged patients, along with diet management and nutrients associated with kidney function. Kidney function in the young and middle-aged population is mostly normal, and they are considered a low-risk group for incident CKD. Thus, we expect this review to be useful in reducing the prevalence of CKD.

## 1. Introduction

Chronic kidney disease (CKD) is a global public health problem that shortens lifespan by increasing the requirement for dialysis treatment and risk of cardiovascular events [1]. The global incidence of CKD was approximately 11 million in 1990 but increased to approximately 21 million (an 88% increase) in 2016. In addition, global deaths due to CKD increased from approximately 600 thousand in 1990 to approximately one million in 2016 [2]. CKD can be caused by immune-related nephropathy [3], polycystic kidney [4], kidney stones [3], renal cell carcinoma [5], or interstitial nephritis/drug-induced kidney dysfunction [6,7]. Furthermore, lifestyle-related diseases, such as obesity [8], hypertension [9], glucose metabolic disorder [10], and dyslipidemia [11,12], are currently important risk factors for CKD. Once CKD occurs, kidney function cannot return to normal; therefore, preventing CKD is important.

Metabolic syndrome is a well-known risk factor for incident CKD [13]; it comprises a high waist circumference (WC) (≥85 cm for men and ≥90 cm for women), hypertension (systolic blood pressure (BP) ≥ 130 mmHg or diastolic BP ≥ 85 mmHg), hyperglycemia (fasting blood glucose ≥ 110 mg/dL), and dyslipidemia (triglycerides (TGs) ≥ 150 mg/dL or high-density lipoprotein cholesterol (HDL-C) < 40 mg/dL). Obesity leads to glomerular hypertrophy, which in turn causes glomerulosclerosis (a scarring in the tiny blood vessels in the glomeruli). A glomerular hypertrophy is an increase in glomerular size and represents both cellular hypertrophy and hyperplasia. To assess obesity, body mass index (BMI), WC, waist-to-hip ratio (WHR), or total body fat percentage (%BF) are measured. Among of them, WC is the most reliable predictor for incident CKD [14]. Obesity, especially abdominal adiposity, is an independent risk factor for incident CKD and estimated glomerular filtration rate (eGFR) decline in men aged 20–60 years [14]. However, in the elderly, %BF was a better predictor of eGFR decline and incident CKD than WC and BMI [15]. Body style and composition change with age; therefore, whether BMI, WC, or %BF is the best predictor of kidney dysfunction might change with the patient’s age.

To modify lifestyle, exercise is recommended, and many people often start to exercise. In metanalysis of 11 randomized controlled trials, exercise provided to favorable effect on eGFR (+2.16 mL/min/1.73 m^2^) compared with placebo (no intervention of exercise) [16]. Furthermore, a renal rehabilitation program was associated with increased survival [17]. Additionally, vigorous intensity activity of 20 min per session 3 days a week is recommended for young individuals [18]. However, it should be noted that a high protein intake results in glomerular hyperfiltration and progresses kidney dysfunction.

Dyslipidemia, a component of metabolic syndrome, is one risk factor for the development of CKD; it is defined as a high level of TG and/or low-density lipoprotein cholesterol (LDL-C) and/or a low level of HDL-C. Both decreased HDL-C and elevated TG levels are associated with the development of CKD [19,20,21]. Furthermore, the TG-to-HDL-C and LDL-C-to-HDL-C ratios were also indicated as predictors of eGFR decline [22,23]. There are several indexes of dyslipidemia for CKD development. Dyslipidemia is a well-known risk factor for atherosclerotic cardiovascular disease (CVD). Lowering LDL-C and TG levels is important to reduce the risk of CVD events according to the clinical guideline by the American heart association [24]. According to the guideline, LDL-C reduction by ≥50% is effective for preventing CVD. In contrast, no specific standard for the management of dyslipidemia to reduce the incidence and progression of CKD has been introduced. Thus, a randomized controlled study should be conducted.

Hypertension is potentially linked with CKD through certain pathophysiological mechanisms. Currently, the therapeutic goal in the management of hypertension in patients with CKD is a resting systolic blood pressure (BP) < 130 mmHg [9]. The recommended drugs for the treatment of hypertension are renin-angiotensin aldosterone system (RAAS) inhibitors. Interestingly, the age-standardized prevalence rate of CKD due to hypertension decreased (by 7.15%) from 1990 to 2016 [2]. Thus, the good management of hypertension in patients with CKD is very important, as it can reduce the risk for CVD events and CKD development.

Hyperglycemia and insulin resistance, the causes of incident diabetes mellitus (DM), are also a serious problem in incident CKD. The age-standardized prevalence of CKD due to DM increased to 9.48% worldwide between and to 2016 [2]. For patients with CKD who have diabetic kidney disease (DKD), eGFR decline is rapid, and it is often difficult to avoid admission for renal replacement therapy. Furthermore, patients with DKD often have CVD, cerebral infarction and hemorrhage, and peripheral arterial disease, which are serious causes of mortality and morbidity. In general, CKD is common in people with DM and hypertension. The combination of DM and hypertension increases in RAAS activity and intraglomerular hypertension, which cause the faster progression of eGFR decline [25].

Young to middle-aged (20 to 60 years old) individuals are not considered a high-risk group for CKD. In addition, there is no management strategy to prevent CKD incidents in them. The early management of kidney dysfunction can improve patient prognosis and reduce mortality and morbidity [26,27]. We believe that many health workers are confused about whether to recommend medical intervention to young and middle-aged individuals with metabolic disorders. Here, this review targets an association between metabolic disorders (excluding hypertension and DM) and the incidence and progression of CKD in young and middle-aged people and is expected to decrease the prevalence of CKD in the future.

## 2. Obesity/Overweight

BMI, WC, and %BF are often used to assess obesity. All of them are well-known predictors of CKD progression [28,29,30]. The prevalence of obesity in patients with CKD increased worldwide, and data from the 2011–2014 National Health and Nutrition Examination Survey (NHANES) showed that 66% of the population surveyed had obesity and 69% had elevated WC [31]. In North America and Europe, obesity is defined as BMI ≥ 30 kg/m^2^, and severe obesity is defined as BMI ≥ 40 kg/m^2^. In contrast, in Asia, BMI ≥ 25 kg/m^2^ defines obesity. In a cohort study of over 60,000 participants, BMI ≥ 25 kg/m^2^ in individuals without any component of metabolic syndrome represented a risk for incident CKD [29].

Elevated BMI has been associated with increased risk for the development of CKD and kidney failure [32,33,34]. In a study of 3.4 million United States (U.S.) veterans with eGFR ≥ 60 mL/min/1.73 m^2^, a BMI between 25 and 30 kg/m^2^ showed the best clinical outcome, and there was a U-shaped curve relationship between BMI and the speed of eGFR decline [33]. By individual meta-analysis including 39 general-population cohorts (mean age, 55 years), compared with BMI of 25 kg/m^2^, the hazard ratio of decline in eGFR for BMI 30, 35, and 40 kg/m^2^ was 1.18, 1.69, and 2.02, respectively [34]. In the middle-aged and older population, elevated BMI might be a risk factor for CKD and eGFR decline. As people age, their body style and composition change; hence, obesity as a risk indicator for CKD might also change.

BMI is simply calculated using body height and body weight, but there are notable limitations. Previous studies focused on BMI as a risk factor for incident CKD; however, it was doubtful whether BMI became a risk factor for incident CKD in the young population (Table 1). BMI does not distinguish body fat from muscle. The cohort, which included middle-aged participants (mean age 45 years old), showed that BMI was not a significant risk factor for incident CKD [35]. In contrast, from the NHANES, abdominal adiposity was significantly related with albuminuria in young, metabolically healthy adults [36]. The prevalence of abdominal adiposity increases after middle-age [37]. Thus, in young people, the assessment of abdominal obesity is important to predict incident CKD.

Among BMI, WC, and %BF, WC was the best predictor of CKD incident in young-to-middled-aged participants [14]. In addition, WC < 80 cm significantly reduced the risk for incident CKD in participants aged <40 years who did not have hypertension or DM, and decreased WC-inhibited eGFR decline [14]. WC was more effective for determining CKD risk than BMI and WHR [38]. However, %BF was more strongly associated with eGFR decline than WC and BMI in the elderly population [15]. Visceral abdominal fat contributes to a higher risk for CKD than subcutaneous fat in the older population cohort (mean age, 74 years) [39].

As mentioned above, abdominal obesity and visceral fat are strongly associated with CKD risk in the young-to-old-aged population. Visceral fat plays a key role in the regulation of numerous cytokines and adipokines [40]. In a cross-sectional study involving 1250 Framingham Heart Study participants, visceral adipose was significantly associated with interleukin-6, *C*-reactive protein, and monocyte chemoattractant protein-1 levels [41]. Furthermore, visceral fat is associated with insulin resistance and DM, which are implied in CKD [42]. Computed tomography is the gold standard for measuring visceral abdominal fat, but it is not appropriate in health screening for the general population due to radiation exposure. Thus, among BMI, WC, or %BF, which are easily measured at health check-ups, WC is the best [39].

We noted that obesity, especially visceral obesity, is a risk factor for CKD, as are hypertension and DM, even in young people. Obesity is a well-known cause of metabolic disorders, CVD events, and higher mortality. It is important to manage body weight from a young age to prevent incident CKD and kidney dysfunction.

## 3. Hypertension

The prevalence of hypertension was 15.5% in Japan and 37.8% in the U.S. from a Japanese cohort of 90,000 and the NHANES, respectively [43]. A gain in BMI of 1 kg/m^2^ increased the odds ratio (OR) for hypertension by 1.23 [43]. Due to the difference in the prevalence of obesity between Japanese and U.S., patients with hypertension are more numerous in Japan [44]. The percentage change in the global prevalence and incidence of CKD due to hypertension decreased from −2.58% in 1990 to −7.15% in 2016. However, that of CKD death due to hypertension increased to 7.33% within the same period [2]. Hypertension leads to the worsening of kidney function and a progressive decline in eGFR, which further impair BP control. Therefore, hypertension should be managed in CKD status and not only in pre-CKD status.

A cohort of about 100,000 Japanese participants showed that the estimated relative risk of end-stage kidney disease (ESKD) increased as BP increased [45]. Arterial hypertension causes endothelial dysfunction and renal microvasculature remodeling [46]. Endothelial dysfunction induces kidney dysfunction and CKD [47]. The blockade of RAAS by angiotensin converting enzyme inhibitors and angiotensin receptor blockers reduces BP and protects microcirculation in the kidney and whole body. CKD is associated with increased RAAS activity. With an upregulation in RAAS activity, blood flow to the glomeruli decreases, further worsening kidney function [9]. Thus, the management of patients with hypertension should commence early, including reducing salt intake, exercise, body weight control, and the use of medications such as RAAS inhibitors. Furthermore, healthy people should aim to prevent hypertension.

Chronic dysfunction in the heart or kidney can cause dysfunction in the other, producing cardio-renal syndrome. Low cardiac output is associated with hypotension and hypoperfusion, which may progress kidney impairment. On the other hand, kidney dysfunction leads to hemodynamic stress, proinflammatory reactions, and arteriosclerotic and cardiomyopathic disease [48]. Importantly, approximately half of patients with heart failure had coexisting chronic kidney dysfunction [48]. Thus, as well as hypertension, the management of heart function is also important for kidney function. However, the prevalence of patients with cardio-renal syndrome was 0.01% (25 to <55 aged) and 0.17% (55 to <65 aged) in the general population [49].

Salt intake is directly related to BP increase and hypertension incidence. Currently, the global usual salt intake ranges between 9 and 12 g/day in the general population, with some differences among the world regions [50]. The World Health Organization (WHO) has established the definition of excessive salt consumption as >5 g/day, which was linked with increased BP and cardiovascular disease events [51]. In contrast, a reduction of salt intake exerted favorable effects, especially decreasing morbidity and mortality associated with cardiovascular disease. The Dietary Approaches to Stop Hypertension diet intervention study, a multicenter study, showed that low salt consumption decreased BP more in women than in men [52]. Thus, it is important for young people to be familiar with a salt-reducing diet so that they can be safe from severe diseases such as CKD and cardiovascular disease events.

Dietary salt intake increases BP; however, salt-sensitive hypertension is different among individuals. Several factors contribute to salt-sensitive hypertension, including environmental condition, aging, mental stress, and genetics [53]. Previous studies have indicated that obesity and female sex are risk factors for salt-sensitive hypertension. The INTERSALT study revealed a significant relationship between the higher excretion of sodium and BP in a cohort of women [54]. Importantly, salt-sensitive hypertension is closely linked with a high level of aldosterone production. Thus, mineralocorticoid receptor blockers are helpful for managing BP in patients with salt-sensitive hypertension.

Hypertension epidemiology in young adults may be different from that in older adults. The prevalence of hypertension was 7.3% among young adults in the U.S. (aged 18–39 years) based on the NHANES database, an increase from 1999 to 2014 [55]. The reason for this increased prevalence might be an increasing prevalence of obesity among young people. Half of young patients with hypertension have not received treatment, and approximately 60% of those who receive treatment do not reach the target BP [56]. From these data, fewer young people visit healthcare clinics and receive therapy for BP control. Therefore, healthcare workers should reach out to young people to help prevent hypertension and provide satisfactory treatment.

The proportion of individuals with hypertension is higher in older cohorts than in the young; therefore, hypertension in the young is often ignored. The increased prevalence of hypertension in the young depended on the increasing prevalence of obesity. Reducing salt intake and controlling body weight will lead to favorable outcomes and decrease the incidence of CKD and cardiovascular diseases in this population. To succeed, healthcare staff need to provide young people with appropriate information regarding hypertension.

## 4. Insulin Resistance/Diabetes Mellitus

DM is a metabolic disorder caused by insulin insensitivity and deficiency. In this section, we discuss type 2 DM (T2DM), which is lifestyle related. This disease becomes a critical health concern as the incidence of CKD, cardiovascular diseases, and aortic atherosclerosis increases, along with the frequency of cancer, infection, and mortality [57,58,59]. In a Chinese cohort of 75,000 people, the prevalence rate of diagnosed T2DM was 12.8%, while that of pre-T2DM was 35.2% [60]. The proportion of patients with T2DM increased steeply after age 50, but 2.0% to 6.3% of patients with T2DM were aged <40 years [60]. The SEARCH study, a five-site study in the U.S., showed that the estimated prevalence of T2DM was 34 cases per 100,000 youth aged 10–19 years in 2001 and 46 cases per 100,000 youth in 2009 (a 31% increase over nine years) [61]. Additionally, the International Diabetes Federation Diabetes Atlas reported that 16% of adults with T2DM were 20 to 39 years old worldwide in 2013 [62]. These increases in T2DM prevalence in the young population have been triggered by of the increasing prevalence of obesity. For young-onset T2DM, obesity is a more common feature than late-onset T2DM. Obesity-related mechanisms include elevated plasma levels of fatty acid and chronic inflammation [63,64]. However, further investigation is required.

DM is the commonest cause of CKD (approximately 40% of newly diagnosed CKD), and the percentage change in CKD incident from 1990 to 2016 was 9.48%, the largest change compared with other causes [2]. Thus, the prevention of T2DM might be tied to the prevention of CKD. DKD, defined as albuminuria > 30 mg/gCr or eGFR < 60 mL/min/1.73 m^2^, was more common in young-onset T2DM [61]. Two cohort studies in Japan and the U.S. indicated that nephropathy caused by T2DM was higher in frequency than that caused by type 1 DM (T1DM) in the young population [61,65]. Furthermore, renal survival was worse in T2DM compared with T1DM. A Canadian study reported that the percentage of patients who were free of ESKD was 100% at 10 years with both T2DM and T1DM, but 92% and 55% at 15 and 20 years with T2DM, respectively, and 100% with T1DM [66]; however, this study had limitations due to its small study size.

Although lifestyle modification is the usual intervention in young-to-middle-aged patients with T2DM, <20% of them maintain or achieve desirable glycemic control with lifestyle modification alone [67]. In contrast, aerobic activity alone or in combination with diet can decrease BP and total cholesterol in overweight patients with young-onset T2DM [68]. From the 2017 Behavioral Risk Factor Surveillance System, the management of CKD accompanied with DM focused on exercise and weight loss predominantly [69]. However, exercise and diet exerted a positive effect on cardiovascular disease risk profile within six months, but not on cardiovascular disease events [67,68].

The pathophysiology of DKD comprises multiple factors, including hyperglycemia, chronic inflammation, increased apoptosis processes, and tissue fibrosis [70]. First, glomerular hypertrophy, which leads to hyperfiltration, occurs in approximately 40% of patients with T2DM [71]. The hyperfiltration of glomeruli progresses to glomerular and interstitial inflammation and the dysregulation of cellular apoptosis [72], thus inducing irreversible damage to nephrons and the progression of kidney dysfunction. Serum creatinine level is normal or low in hyperfiltration. Unfortunately, most patients with T2DM are not yet aware that they have DKD. Although DKD is not detected by serum creatinine levels, abnormal urine albumin is useful in the detection of DKD. DKD can traditionally be identified by the presence of an elevated urine albumin/creatinine ratio (UACR, ≥30 mg/g) [73]. However, 63.7% of patients with T2DM in the United Kingdom National Diabetes Audit had decreased eGFR without elevated albuminuria [74]. Thus, a novel and useful biomarker for the early detection of DKD is desired.

Regarding the relationship between young-onset T2DM and DKD, obesity plays a key role in incident T2DM in young and middle-aged patients. Therefore, body weight control by diet and aerobic activity might reduce the incidence of T2DM. The measurement of serum creatinine levels or UACR at regular health check-ups is promising for the early detection of kidney dysfunction.

## 5. Dyslipidemia

Hyperlipidemia, a high level of LDL-C and TGs, potentially accelerates the progression of eGFR decline by several mechanisms. There are several previous reports on the association between dyslipidemia and kidney dysfunction (Table 2). First, the reabsorption of fatty acid cholesterol contained in excretion proteins in epithelial cells induces interstitial inflammation and tissue damage [75,76]. Second, lipoprotein accumulation in the glomerular matrix promotes matrix expansion and glomerular sclerosis [77,78]. Additionally, oxidized lipids injure the glomeruli and worsen kidney dysfunction [79]; they also cause oxidative stress and injury to epithelial cells, promoting glomerulosclerosis [80]. Dyslipidemia is a well-known risk factor for atherosclerosis and cardiovascular disease; therefore, renal aortic atherosclerosis secondary to dyslipidemia might cause CKD. Thus, as mentioned above, changes in the lipid profile can directly damage glomeruli and interstitial tissue.

Dyslipidemia implies a high level of serum TG or LDL-C or a low level of serum HDL-C. Dyslipidemia is a risk factor for the development of CKD [11,12] and cardiovascular events [11,81,82]. The working group of Kidney Disease: Improving Global Outcomes in 2013 suggests statin treatment in adults aged 18–49 years who have CKD and have previously received renal replacement therapy [83]. However, the working group did not set the target level of LDL-C during treatment using statin. In patients with CKD who have DM, the occurrence rate of incident myocardial infarction or coronary death exceeds 10 per 1000 patient-years. Similarly, cardiovascular risk is high even in patients with CKD aged <50 years who do not have DM or prior coronary events. The 10-year incidence rate of coronary events or death may be assessed using several evaluation tools such as the Framingham risk score [84], Systemic Coronary Risk Evaluation Project [85], ASSIGN score in Scottish heart health extended cohort [86], or the prospective cardiovascular Munster study [87]. Statin treatment is somewhat harmful in higher LDL-C; however, it is effective in reducing the 10-year risk of coronary death or non-fatal myocardial infarction.

High levels of LDL-C reduce eGFR in participants with normal kidney function by impairing the function of renal arterioles and glomeruli capillaries [88]. A retrospective cohort study of 15,000 workers aged 20–60 years with basal eGFR ≥ 60 mL/min/1.73 m^2^ reported that a serum LDL-C level > 140 mg/dL was a significant risk factor for incident CKD (defined as eGFR under 60 mL/min/1.73 m^2^) at the 5-year follow-up [23]. Interestingly, the cut-off was appropriate for indicating incident CKD in workers with and without hypertension who did not have DM, but not in those who had DM. In contrast, the change in eGFR at 5-year follow-up decreased in the group with higher basal LDL-C levels than in the lower LDL-C group [23]. The Justification for the Use of Statin in Prevention: An Intervention Trial Evaluation Rosuvastatin study showed that rosuvastatin therapy attenuated eGFR decline after one year compared with placebo in an apparently healthy population with basal eGFR 60 mL/min/1.73 m^2^ [89]. However, there was a curiously inverse association between LDL-C levels and kidney disease outcomes (ESKD or 50% decline in eGFR) in patients with CKD who had low levels of proteinuria [90]. The reason for this paradoxical relationship remains unclear.

TG levels were inversely associated with eGFR in middle-aged and elderly Chinese individuals with basal eGFR ≥ 90 mL/min/1.73 m^2^ [22]. The OR for decline in eGFR to 60–90 mL/min/1.73 m^2^ was 1.61 in the highest TG quartile group. In a cross-sectional study involving 9100 individuals in China, the ORs for predicting an eGFR of 60–90 mL/min/1.73 m^2^ were 1.39 and 1.19 in individuals aged 18–45 years and 45–65 years, respectively [91]. The Atherosclerosis Risk in Communities study reported that the relative risk for a 0.4 mg/dL increase in serum creatinine levels for the highest and lowest TG quartile was 1.65 (95% confidence interval (CI), 1.1–2.5) in participants with normal serum creatinine levels [20]. However, that study did not mention a significant risk of increased serum creatinine for LDL-C.

HDL-C is the primary mechanism of retrieval and transport of surplus cholesterol from the extrahepatic area into the liver [92]. This process protects against atherosclerosis and kidney dysfunction. A prospective cohort study involving 4483 apparently healthy men indicated that the multivariable-adjusted relative risks for elevated serum creatinine (≥1.5 mg/dL) were 2.12 (95%CI, 1.39–3.22) and 2.22 (95%CI, 1.27–3.89) for HDL-C < 40 mg/dL and the ratio of total cholesterol/HDL-C ≥ 6.8, respectively [93]. Most of the study’s participants were younger and had normal kidney function (mean age 48.5 years) and only 3.0% of them had serum creatinine levels > 1.5 mg/dL. HDL-C alone as predictor for CKD have been less reported. Many studies used the TG/HDL-C or total cholesterol/HDL-C ratio.

The TG/HDL-C ratio has been considered a useful indicator of insulin resistance [94,95]. Additionally, this ratio was inversely correlated with eGFR decline. A Japanese longitudinal cohort study involving 124,700 individuals out of the general population indicated that the OR of the highest quartile of TG/HDL-C ratio versus lowest quartile was 1.25 (95%CI, 1.18–1.34) for CKD at 2-year follow-up [81]. The study also showed that a higher TG/HDL-C ratio can imply a risk of eGFR decline and incident proteinuria. Conversely, the total cholesterol/HDL-C ratio showed a 50% higher risk for eGFR < 60 mL/min/1.73 m^2^ with each standard division increment in a cross-sectional study involving 11,956 individuals out of the general population aged ≥35 years in China [96].

The LDL-C/HDL-C ratio is recognized as a potential indicator of the progression of cardiovascular diseases and atherosclerosis [97,98]. However, there is less evidence for LDL-C/HDL-C ratio as an indicator for CKD and eGFR decline. In participants with hypertension aged 20–60 years, the LDL-C/HDL-C ratio was a significant indicator for CKD (<60 mL/min/1.73 m^2^) at 5-year follow-up [23]. Furthermore, the area under the receiver operating characteristic (ROC) curve of the LDL-C/HDL-C ratio for identifying eGFR < 60 mL/min/1.73 m^2^ was the smallest among the total cholesterol/HDL-C, TG/HDL-C, and LDL-C/HDL-C ratios [96]. Thus, although the LDL-C/HDL-C ratio might be a poor predictor for kidney dysfunction, it is a fine indicator for cardiovascular diseases and atherosclerosis.

The prevalence of dyslipidemia has increased due to changing diet and lifestyle and an increasingly overweight population. There has been a lot of evidence for cardiovascular diseases and atherosclerosis but not for the incidence and progression of CKD. More studies are required to establish good evidence and prevent kidney dysfunction in the young and middle-aged.

## 6. Hyperuricemia

Hyperuricemia, that is, elevated serum uric acid (SUA), is a lifestyle-related disease, but most patients with hyperuricemia have no symptoms, which is worrisome. Some studies recommend lower SUA values < 5 mg/dL for the apparently healthy population, while the European League Against Rheumatism recommends a target of <6.0 mg/dL with medication for anti-hyperuricemia [99,100,101]. In contrast, symptomatic hyperuricemia comprises gout attacks, urolithiasis, and uric acid nephropathy [102]. Uric acid is produced by the metabolism of both endogenous (daily approximate 300–400 mg) and exogenous (daily approximate 300 mg) purines, about 70% of which are excreted in urine [103]. Thus, SUA increases with decreasing eGFR.

A prospective study involving over 20,000 healthy individuals showed that an increase in SUA from 7.0 to 8.9 mg/dL was associated with a two-fold increase in the risk for incident CKD [104]. Similarly, another study involving 13,338 healthy individuals with normal kidney function found that the OR for eGFR < 60 mL/min/1.73 m^2^ increased by 1.1 for every 1.0 mg/dL increase in SUA after adjustments for several metabolic parameters [105]. Thus, previous studies indicated an association between elevated SUA levels and CKD. However, there is still no consensus on an approximate cut-off value of SUA for the prediction and prevention of CKD. In healthy men with baseline SUA < 7.0 mg/dL, an increase to >7.0 mg/dL significantly reduced eGFR at 5-year follow-up compared with SUA remaining at <7.0 mg/dL [106]. Furthermore, the appropriate cut-off value of SUA for CKD was reported to be 6.6 mg/dL in both young and middle-aged men [107]. That study. included 8207 healthy participants with normal kidney function, observed CKD incidence at 5-year follow-up, and performed analysis after propensity score matching [107]. However, the area under the ROC curve for SUA was small. Thus, evidence for a target SUA value to predict CKD are still needed.

Some countries, such as Japan, recommend treatment for asymptomatic hyperuricemia. One study indicated that hyperuricemia was linked with an increase in the risk for hypertension, dyslipidemia, CKD, and weight gain [108]. SUA-lowering therapy is associated with reduced eGFR decline. A randomized clinical trial in Spain showed that allopurinol treatment reduced the progression of kidney disease and that the mean SUA of participants was 7.8 mg/dL [109]. Furthermore, reduction in SUA to <7.0 mg/dL significantly decreased eGFR decline at 5-year follow-up in 1642 individuals from the general population [106]. In contrast, a recent systematic review including 13 randomized controlled trials recommended that asymptomatic hyperuricemia be treated only under certain conditions: persistent SUA levels > 13 mg/dL in men or >10 mg/dL in women, >1100 mg daily urinary excretion of uric acid, or before the administration of radiation or chemotherapy [110]. However, five of the 13 trials observed for changes in eGFR or serum creatinine, whereas the others observed for only kidney function, cardiovascular diseases, and SUA-lowering by medication.

The risk for CKD with higher SUA differs between young and middle-aged men. Higher SUA levels (≥6.6 mg/dL) were not a risk factor for CKD in young men who are not obese (BMI < 25 kg/m^2^), but there was a significant risk factor for CKD in young men with BMI ≥ 25 kg/m^2^ (OR = 2.18, 95%CI, 1.10–4.31) [107]. Additionally, there was a significant interaction between higher SUA levels and BMI. However, higher SUA was significantly associated with CKD (OR = 1.32, 95%CI, 1.07–1.63) in middle-aged men regardless of BMI [107]. The prevalence of abdominal adiposity is higher in middle-aged than in young men even when BMI is under 25 kg/m^2^ [37]. A higher SUA accompanied by obesity (especially abdominal adiposity) might be associated with CKD but high SUA levels alone are not.

Lifestyle modification has been a good treatment for hyperuricemia and gout [111]. To decrease SUA, the rheumatology society guidelines recommend weight loss, reducing the intake of purine-rich foods, and reducing the consumption of alcohol and fructose-containing juice [112]. A systematic review of 10 articles showed that weight loss had a favorable effect on lowering SUA and achieving target SUA levels [113]. However, weight loss and dietary modification are difficult to maintain in the long term, and there is insufficient evidence for the benefit of lifestyle modification.

## 7. Diet and Nutrients

There is still no practical evidence for the efficacy of diet therapy for the prevention of CKD; however, a diet for reducing obesity/overweight, hypertension, DM, dyslipidemia, and hyperuricemia can produce better results for kidney function. The Kidney Disease Outcomes Quality Initiative (KDOQI) guidelines updated in 2020 recommended the nutritional management of patients with CKD [114]. In this section, we discuss diet management focused on salt, protein, and calories.

Sodium intake impacts the progression of CKD and cardiovascular disease. High sodium intake induces whole-body volume overload and hypertension, which implies CKD progression and cardiovascular remodeling [115,116,117]. Conversely, interventional studies with sodium intake restriction indicated decreasing BP and proteinuria, which helped to attenuate CKD progression and mortality [118]. According to WHO recommendations, restricting sodium intake to <2.3 g/day (5.8 g/day of salt) is the most cost-effective measure in public health [51].

In contrast, the evaluation of low salt consumption is an important concern. The measurement of urinary sodium excretion over 24 h (UNaV) is considered the gold standard for evaluating salt consumption. The Chronic Renal Insufficiency Cohort study, a longitudinal prospective study involving 3757 patients with CKD (mean eGFR, 43.4 mL/min/1.73 m^2^), showed that the highest quartile of UNaV was significantly associated with a high risk for ESKD [119]. However, another study including non-CKD participants (mean eGFR, 68.4 mL/min/1.73 m^2^) did not indicate an association between UNaV and kidney outcomes (ESKD or eGFR decline). In that study, the post hoc analysis of the Ongoing Telmisartan Alone and in Combination with Ramipril Global Endpoint Trial and the Telmisartan Randomized Assessment Study in ACE-Intolerant Subjects with Cardiovascular Disease study, the ORs for low (median, 3.3 g/day) and high (median, 6.1 g/day) sodium excretion for a 30% decline in eGFR and the progression of proteinuria were 0.99 (95%CI, 0.89–1.09) and 0.92 (95%CI, 0.84–1.01), respectively [120]. Thus, an appropriate evaluation of salt intake in normal kidney function remains to be performed.

For patients with CKD, the recommended protein intake is 0.55 to 0.60 g/kg/day according to the 2020 KDOQI guidelines [114], but there is no recommendation for protein restriction for the general population without CKD. Protein is broken down into amino acids to produce energy. Unfortunately, excess protein intake cannot be stored within the body. In contrast, high nitrogenous waste products degraded from protein are harmful. Thus, excess protein intake might be undesirable, even when kidney function is normal.

A high load of amino acids expands the afferent arteriole and increases intraglomerular pressure, resulting in glomerular hyperfiltration. Conversely, a low protein diet ameliorates afferent arteriole expansion and suppresses high glomerular filtration load [121]. Several meta-analyses showed that protein restriction produced favorable outcomes for CKD development, BP, and proteinuria [122,123,124]. Although there is still no recommended standard of protein intake to prevent CKD in the general population, unnecessary protein consumption, including amino acid supplements, should be avoided.

A high-calorie diet induces overweight/obesity, and obesity causes CKD onset and progression [28,29,30]. As described above, obesity is an independent risk factor for hypertension, T2DM, dyslipidemia, and hyperuricemia. Exercise and calorie restriction reduce oxidative stress and chronic inflammation [125]. In addition, weight loss leads to obesity-related glomerular hyperfiltration and albuminuria [126,127]. A 2-year follow-up study involving a young and middle-aged population showed that 25% calorie restriction for two years reduced systolic and diastolic BP, glucose tolerance, LDL-C levels, and *C*-reactive protein levels [128]. These data support appropriate calorie intake to contribute to kidney function.

The long chain omega-3 polyunsaturated fatty acids (LC *n*-3 PUFA), which are derived from fish oil, include eicosapentaenoic (EPA), docosapentaenoic, and docosohexaenoic acids (DHA). The LC *n*-3 PUFA significantly lowered the CVD events rate (relative risk, 0.41 versus corn-oil-based placebo) [129], but there is insufficient evidence of LC *n*-3 PUFA for eGFR decline. Based on six randomized controlled trials with CKD stage 3 populations, fish oil supplementation was not found to influence eGFR [114]. In most of those trials, participants were received 1.8 g/day of LC *n*-3 PUFA [130]. Furthermore, the daily intake of 1.8–3.6 g of LC *n*-3 PUFA resulted in decreased TG, increased HDL-C, and a reduction in interleukin-6 levels, which can provide favorable effects for kidney function [130]. These data support the daily intake of fish oil for CKD, but further studies are expected.

## 8. Conclusions

Most young and middle-aged individuals have normal eGFR (>60 mL/min/1.73 m^2^) and are not considered a high-risk group for CKD. However, the prevalence of patients with CKD as well as obesity, hypertension, and T2DM has increased due to changes in lifestyle. Once kidney dysfunction occurs, kidney function cannot recover to the original (normal) state. Thus, it is extremely important to prevent CKD from developing in young and middle-aged populations. In this review, we focused on lifestyle-related risk factors for the onset and progression of CKD. We believe that dietary habits and body weight control are important for suppressing lifestyle-related diseases (especially dyslipidemia and hyperuricemia) and preventing CKD. Based on this review, we suggest the target range of good control of metabolic factors for CKD management as follows: BMI < 25 kg/m^2^ for Asian and <30 kg/m^2^ for non-Asian populations, BP < 130/80 mmHg [131], daily salt intake < 5 g, hemoglobin A1c ≤ 7% [132], albuminuria < 30 mg/g, LDL-C < 140 mg/dL, and lowering the ratio of TC/HDL-C and serum level of urate < 7.0 mg/dL. However, there is less evidence for lifestyle-related disease prevention through favorable lifestyle habits for kidney function in the young and middle-aged. More studies on this are required in the future.

## Figures and Tables

**Table 1 nutrients-14-03787-t001:** Outcomes of previous studies focused on risk of obesity for chronic kidney disease in young- and middle-aged people.

Author, year	Risk Factor	Number	Age	Study Design	Study Outcomes	Reference
Elsayed E, et al., 2008	BMI ≥ 27.2 kg/m^2^, WHtR ≥ 0.96 (men), ≥0.89 (women)	13,324	mean 57.4 years	cohort study, over 9.3-year follow-up	BMI; OR 0.99 (95%CI 0.96–1.03) for CKD incident *^1^ WHtR; OR 1.17 (95%CI 0.99–1.34) for CKD incident *^1^	[28]
Noori N, et al., 2009	WC, WHR	3107	mean 40 years	cohort study, 7-year follow-up	WC (highest quartile); HR 1.88 (95%CI 1.17–3.01) for CKD incident *^2^ WHR; no association between WHR and CKD incident	[38]
Vivante A, et al., 2012	BMI	1,194,704	mean 17.4 years	cohort study, enrolled from 1967 to 1997 and incident cases between 1980 to 2010.	overweight (85th–94th in BMI); HR 3.00 (95%CI 2.50–3.60) and obese (≥95th in BMI); HR 6.89 (95%CI 5.52–8.59) for ESKD	[32]
Song YM, et al., 2015	BMI ≥ 25 kg/m^2^	1881	mean 43.9 years	cohort study, 3.7-year follow-up	OR 2.03 (95%CI 1.05–3.92) for CKD incident *^2^	[30]
Lu JL, et al., 2015	BMI	3,376,187	≥20 years	cohort study, 7-year follow-up	No association in under 40 years old participants. BMI displayed a U-shaped association with eGFR decline (>5 mL/min/1.73 m^2^) and BMI 25–30 kg/m^2^ was lowest risk for eGFR decline in over 40 years old participants.	[33]
Hashimoto Y, et al., 2015	BMI	3136	mean 45.3–52.2 years	cohort study, 8-year follow-up	OR 0.83 (95%CI 0.36–1.72) in metabolic healthy participants. OR 2.80 (95%CI 1.45–5.35) in metabolically abnormal participants for incident CKD *^2^	[35]
Dai D, et al., 2016	BMI ≥ 25 kg/m^2^, WC ≥ 84 cm (men), ≥81 cm (women), WHtR ≥ 0.5	11,192	mean 53.83 years	cross section	BMI; OR 2.27 (95%CI 1.06–4.82) (men), OR 1.80 (95%CI 1.04–3.10) (women) for CKD incident *^2^ WC; OR 1.75 (95%CI 0.97–3.15) (men), OR 2.12 (95%CI 1.25–3.58) (women) for CKD incident *^2^ WHtR; OR 3.20 (95%CI 1.28–7.95) (men), OR 1.87 (95%CI 1.07–3.25) (women) for CKD incident *^2^	[15]
Chang Y, et al., 2016	BMI ≥ 25 kg/m^2^	62,249	mean 36.1 years	cohort study, 369,088 person-year follow-up	OR 6.7 (95%CI 3.0–10.4) for CKD incident *^2^	[29]
Sarathy H, et al., 2016	WC ≥ 102 cm (men), ≥88 cm (women)	6913	20–40 years	cohort study, 10-year follow-up	OR 3.0 (95%CI 1.7–5.4) for albuminuria *^3^ in Mexican-Americans	[36]
Kuma A, et al., 2019	WC ≥ 80 cm (men)	8015	20–60 years	cohort study, 5-year follow-up	OR 1.57 (95%CI: 1.35–1.84) for CKD incident *^4^	[14]

BMI: body mass index, WHtR: waist-to-height ratio, WC: waist circumference, WHR: waist-to-hip ratio, OR: odds ratio, CI: confidence interval, HR: hazard ratio. Definition of CKD: *^1^: eGFR < 60 mL/min/1.73 m^2^ or decline in eGFR, *^2^: eGFR < 60 mL/min/1.73 m^2^, *^3^: ≥30 mg/g creatinine, *^4^: eGFR < 60 mL/min/1.73 m^2^ and proteinuria with dipstick ≥ 1+.

**Table 2 nutrients-14-03787-t002:** Outcomes of previous studies focused on risk of dyslipidemia for chronic kidney disease in young- and middle-aged people.

Author, Year	Risk Factor	Number	Age	Study Design	Study Outcomes	Reference
Schaeffner ES, et al., 2003	ratio of TC/HDL-C ≥ 6.8	4483	mean 48 years	cohort study, 14-year follow-up	RR 2.22 (95%CI 1.27–3.89) of elevated serum creatinine level (≥1.5 mg/dL)	[93]
Rahman M, et al., 2014	TC, LDL-C	3939	mean 58.2 years	cohort study, 4.1-year follow-up	Not significant association between dyslipidemia and 50% decline in eGFR or ESKD.	[90]
Hou X, et al., 2014	TC, TG, HDL-C, LDL-C	2647	≥40 years	cross section	OR 1.61 (95%CI 1.12–2.32) in highest quartile of TG for mildly reduced eGFR *^1^. Not significant risk of TC, LDL-C, and HDL-C.	[22]
Tsuruya K, et al., 2015	ratio of TG/HDL-C > 3.02 (men), >2.20 (women)	102,900	≥40 years	cohort study, 2-year follow-up	OR 1.25 (95%CI 1.18–1.34) of incident CKD *^2^	[81]
Kuma A, et al., 2018	LDL-C ≥ 140 mg/dL	14,510	20–60 years	cohort study, 5-year follow-up	OR 1.46 (95%CI 1.12–1.90) without hypertension, and OR 1.49 (95%CI 1.23–1.82) without DM for incident CKD *^3^	[23]
Wang H, et al., 2018	ratio of TC/HDL-C ≥ 3.07, TG/HDL-C ≥ 0.62, LDL-C/HDL-C > 2.64	3259	mean 59 years	cross section	OR 2.85 (95%CI 1.23–6.25) of TC/HDL-C, OR 3.96 (95%CI 1.58–9.92) of TG/HDL-C, OR 2.22 (95%CI 1.15–4.29) of LDL-C/HDL-C for incident CKD *^3^	[96]
Xue N, et al., 2019	TG, HDL-C, LDL-C	9100	18–65 years	cross section	TG: OR 1.17 (95%CI 1.07–1.29), HDL-C: OR 0.54 (95%CI 0.38–0.76) for early eGFR decline *^1^. Not significant risk of LDL-C.	[91]

TC: total cholesterol, HDL-C: high-density lipoprotein cholesterol, LDL-C: low-density lipoprotein cholesterol, RR: relative risk, CI: confidence interval, OR: odds ratio. Definition of CKD: *^1^: eGFR = 60–90 mL/min/1.73 m^2^, *^2^: eGFR < 60 mL/min/1.73 m^2^ and/or proteinuria with dipstick ≥ 1+, *^3^: eGFR < 60 mL/min/1.73 m^2^.

## Data Availability

Not applicable.

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
