# Peer review of "Lifestyle-Related Risk Factors for the Incidence and Progression of Chronic Kidney Disease in the Healthy Young and Middle-Aged Population"

_nutrients, 2022, doi:10.3390/nu14183787_

Round 1

Reviewer 1 Report

The review evaluates the risk factors like obesity, hypertension, diabetes mellitus, dyslipidemia, and hyperuricemia for the young and middle-aged populations. The review is concise and well-written. I suggest it be accepted with a couple of minor changes. 

  1. The introduction section of the review introduces some terms that should be explained and defined for readers that are not familiar with them. For eg. glomerular hypertrophy, and glomerulosclerosis.  
  2.  The authors should cite the paper by Hongfang Liu (Detecting lifestyle risk factors for chronic kidney disease with comorbidities: association rule mining analysis of web-based survey data; J. Med Internet Res 2019; 21(12):e14204); doi: 10.2196/14204) that concludes that the management of CKD patients with diabetes focuses on exercise and weight loss predominantly.

Author Response

  1. We appreciate for your suggestion. A glomerular hypertrophy is an increase in glomerular size, represents both cellular hypertrophy and hyperplasia. And, a glomerulosclerosis is a scarring in the tiny blood vessels in the glomeruli. We added explanation of them. (page 1 line 42-44)
  2. We appreciate for your providing good reference. We cited the literature and added sentence. (page 6 line 251-252).

Reviewer 2 Report

Dear Editor and authors

This is a thorough review of the factors related to metabolic syndrome and exacerbation of chronic kidney disease (CKD).

"Lifestyle-related Risk Factors for the Incidence and Progression of Chronic Kidney Disease in the Healthy Young and Middle aged Population"

It is a complete and detailed review of related topics of hypertension (HTN), diabetes (DM), hyperlipidemia, hyperuricemia, obesity and progression of CKD.

Several points for further explanation may be considered.

1.      Exercise may be related to obesity and is important to life style modification.

Stationary life style, regular exercise, or heavy exercise (with probably high protein intake) may contribute to different outcome of CKD.

2.      As DM and HTN contribute to more than half of the newly found uremic patients, the combining effect of HTN with DM may need more explanation because comorbidities may contribute to faster decline of renal function.

3.      The heart function may be mentioned in the HTN section due to poor heart function may also contribute to poor renal function, the cardio-renal syndrome. If the percentage is low in the target population (young and middle aged) please state it.

4.      In the diet section for CKD, the benefit effect of fish oil is still in debate. How fish oil may be helpful in certain kidney diseases can be added in this paragraph if applicable.

5.      If possible, please provide the target range of good control of DM, HTN, hyperlipidemia, hyperuricemia and obesity for this young and middle aged population and add to the conclusion. This may help clinical doctors easily catch the points provided by this article after being persuading the importance to control these risk factors.

Only minor revision is required with English polishing. Thanks for contributing this interesting review.

Author Response

  1. We agree with your comment. Exercise is so important for kidney function. In metanalysis of 11 RCTs, exercise provided favorable effect on eGFR (+2.16 ml/min/1.73m2) compared with placebo. Furthermore, renal rehabilitation is associated with increased survival. The vigorous activity (20 minuets per session, 3 days a week) was recommended for younger individuals. However, a high protein intake results in glomerular hyperfiltration, which may provide kidney dysfunction. We added them in introduction section. (page 2 line 52-58)
  2. We agree with your comment. CKD is common in people with diabetes mellitus and hypertension. The combination of diabetes and hypertension increases in renin-angiotensin system activity and intraglomerular hypertension, which progress faster eGFR decline. We added sentences in introduction section. (page 2 line 85-87)
  3. Thank you for good suggestion. According to your comment, we added contents of cardio-renal syndrome in the hypertension section. (page 5 line 175-183)
  4. Thank your for suggestion. According to your comment, we wrote the association between fish oil and CKD in Diets and Nutrients section. (page 11 line 471-480)
  5. We appreciate your suggestion. Based on this review, we suggest the target range of good control of some metabolic parameters for CKD management as follows: BMI <25 kg/m2 for Asian and <30 kg/m2 for except Asian, BP <130/80 mmHg, daily salt intake <5g, hemoglobin A1c <7%, albuminuria <30mg/g, LDL-C <140 mg/dL, lower the ratio of TC/HDL-C, and serum level of urate <7.0 mg/dL. We added sentence in conclusions section. (page 11 line 490-494)